# Hedging Against Neuropathic Pain: *Role of Hedgehog Signaling in Pathological Nerve Healing*

**DOI:** 10.3390/ijms21239115

**Published:** 2020-11-30

**Authors:** Nathan Moreau, Yves Boucher

**Affiliations:** 1Department of Oral Medicine and Oral Surgery, Bretonneau Hospital (AP-HP), 75018 Paris, France; nathan.moreau@u-paris.fr; 2Faculty of Dental Medicine-Montrouge, University of Paris, 92120 Montrouge, France; 3Department of Dental Medicine, Pitié-Salpêtrière Hospital (AP-HP), 75013 Paris, France; 4Faculty of Dental Medicine-Garancière, University of Paris, 75006 Paris, France

**Keywords:** sonic hedgehog, desert hedgehog, peripheral nerve injury, nerve healing, neuropathic pain

## Abstract

The peripheral nervous system has important regenerative capacities that regulate and restore peripheral nerve homeostasis. Following peripheral nerve injury, the nerve undergoes a highly regulated degeneration and regeneration process called Wallerian degeneration, where numerous cell populations interact to allow proper nerve healing. Recent studies have evidenced the prominent role of morphogenetic Hedgehog signaling pathway and its main effectors, Sonic Hedgehog (SHH) and Desert Hedgehog (DHH) in the regenerative drive following nerve injury. Furthermore, dysfunctional regeneration and/or dysfunctional Hedgehog signaling participate in the development of chronic neuropathic pain that sometimes accompanies nerve healing in the clinical context. Understanding the implications of this key signaling pathway could provide exciting new perspectives for future research on peripheral nerve healing.

## 1. Introduction

Contrary to its central counterpart, the peripheral nervous system exhibits a high regenerative capacity notably following peripheral nerve injury [1,2]. Such regenerative capacity is thought to result from the remarkable plasticity of Schwann cells, key effectors of peripheral nerve healing and homeostasis [3,4]. Following injury, Schwann cells transdifferentiate into repair Schwann cells that express a new genetic repair program to allow nerve healing and the restoration of peripheral nerve homeostasis [2]. This new repair program includes the activation of proregenerative genes that are repressed during development and in uninjured nerves such as the morphogen Sonic Hedgehog [2,5,6].

Hedgehog signaling plays an essential morphogenetic role in the development of the central [7,8,9,10] and peripheral nervous system [11], with major roles in patterning and cell fate specification [9]. The presence and/or reactivation of morphogens in adulthood has shed light on the implication of such morphogenetic pathways in central nervous system homeostasis [9,12] and disease [13] but also in physiological [14,15,16] and pathological peripheral nerve healing [16,17] and in neuropathic pain development [18,19,20].

This review aimed to summarize the current data regarding the roles of Hedgehog morphogens and underlying signaling pathway in peripheral nerve morphogenesis, physiological and pathological nerve healing, and neuropathic pain development. Future research and clinical perspectives are also discussed.

## 2. Hedgehog Signaling in Peripheral Nerve Morphogenesis

The Hedgehog pathway, roughly comprised of three proteins Sonic Hedgehog (SHH), Desert Hedgehog (DHH), and Indian Hedgehog (IHH), three transmembrane receptors (Patched-1, Patched-2, and Smoothened), and three transcription factors (Gli-1, Gli-2, Gli-3), is essential for the ontogenic development of mammals [9] with several time-dependent and concentration-dependent morphogenetic effects on numerous cell lines, including the central [12] and peripheral nervous system [11].

Whereas the role of Hedgehog morphogens (and underlying signaling pathway) has been extensively studied in embryonic development and in the patterning/differentiation of the central nervous system (e.g., dorsal–ventral axis patterning of the neural tube and establishment of distinct ventral neuronal populations in a concentration-dependent manner) [9], the implication of the Hedgehog pathway is less clear in the peripheral nervous system. For instance, SHH is not expressed in Schwann cells at any stage of development [5,6] (as it is constitutively repressed during embryonic and promyelinating states of development by the polycomb pathway [4]) whereas DHH plays key roles in peripheral nerve structure specification during development [11].

During peripheral nerve morphogenesis the concomitant and symbiotic development of both neuronal axons and Schwann cells drive the formation of the myelin sheath and specialization of axonal structures such as the nodes of Ranvier [21]. Such axo-glial interactions are under the dependence of Schwann cell development, originating from neural crest cells that differentiate into Schwann cell precursors, then into immature Schwann cells before specification into myelinating or nonmyelinating Schwann cells [21]. Myelinating Schwann cells form a 1:1 contact with large caliber axons, wrapping them in a myelin sheath whereas nonmyelinating Schwann cells (also called Remak Schwann cells) surround multiple small caliber axons without any myelination [2]. Schwann cells and axons are supported and insulated by a connective tissue sheath of fibroblasts scattered between the nerve fibers that form the epineurium, perineurium, and endoneurium of the peripheral nerve [22].

The appearance of immature Schwann cells during development appears to coincide with the generation of the peripheral nerve’s extracellular matrix and the development of endoneurial connective tissue and blood vessels [23]. DHH and vascular endothelial growth factor (VEGF)-mediated signaling, originating from such immature Schwann cells and related axons, drive the differentiation and organization of arterial and perineurial cells, modeling the peripheral nerve’s architecture [11].

Indeed, deletion of the *Dhh* gene in mice results in the disruption of the fascicular structure of peripheral nerves, with a thin and disorganized perineurial sheath and an increase in blood–nerve barrier permeability [11]. Interestingly, a similar structural anomaly was found in a case of homozygous missense mutation of the *dhh* gene in a human diagnosed with “minifascular neuropathy” [24].

## 3. *Shh*aping Peripheral Nerve Healing and Homeostasis

Following peripheral nerve injury (such as a nerve transection), myelinating and nonmyelinating Schwann cells undergo extensive reprogramming, to promote and guide axonal repair [2]. Schwann cells lose contact with the distal axonal stump, demyelinate the stump (following a physiological process called Wallerian degeneration), and then develop a repair phenotype (becoming repair Schwann cells), involving notably the downregulation of several promyelinating genes and the de novo expression of other regeneration genes [2]. Such Schwann cell reprogramming involves the upregulation of several genes and subsequent activation of multiple transcriptional mechanisms that control and regulate said repair program, including c-Jun, mitogen-activated protein kinase (MAPK) pathways, and SHH [2,4,5].

Repair Schwann cells allow the disintegration and removal of damaged axons during Wallerian degeneration and assist myelin debris clearance, thus creating a favorable environment for nerve regrowth. Such myelin clearance is achieved by clearing both intrinsic and extrinsic myelin fragments by myelinophagy (i.e., Schwann cell autophagy [25]) and phagocytosis, notably by recruited and activated blood-borne macrophages [2,26].

Afterwards, Schwann cells secrete trophic factors to support survival of damaged neurons and promote axon regrowth [2]. Repair and Remak Schwann cells also extent long parallel processes, aligned in tracts, called bands of Büngner. Such bands serve to guide the regrowing axon back to innervate its initial target [5]. Finally, Schwann cells proliferate, upregulate promyelinating genes, redifferentiate into myelinating Schwann cells, and remyelinate the regenerated axon [2].

Historically, the phenotypic transformation of Schwann cells following peripheral nerve injury has been considered a form of dedifferentiation, mirroring the phenotype of immature Schwann cells during peripheral nerve development [27,28], notably the repression of myelin genes [3,4] secondary to the remodeling of the Schwann cell epigenome [29]. However, denervated Schwann cells have additional functions such as axonal regeneration [4], myelin degradation or recruitment of macrophages [6,27], and activate genes never before expressed during development (and thus unique to the injury condition) such as *Shh* [5,6,30], suggesting that peripheral nerve injury does not induce Schwann cell dedifferentiation (i.e., returning to a previous developmental stage) but transdifferentiation (i.e., evolving into a different phenotype).

### 3.1. The Multifaceted Role of Hedgehog Signaling during Peripheral Nerve Healing

The study of nerve regeneration in peripheral nerve injury models has highlighted the role of morphogens in peripheral nerve healing, notably as part of the Schwann cell-driven repair program [4]. Indeed, as *Shh* is expressed immediately after a crush injury to the sciatic nerve [15,31], it has been suggested that Hedgehog signaling could likely play an important role in the recovery of injured nerves.

Current scientific evidence, summarized herein, illustrates the previously unsuspected multifaceted role of Hedgehog signaling during peripheral nerve healing.

#### 3.1.1. Promotion of Axonal Survival

SHH could promote axonal survival following peripheral nerve injury [15,16,32], by promoting the survival of injured neurons [14] or via a neurotrophic effect [15] (see Section 3.1.4), as similarly observed in the central nervous system following oxidative stress or oxygen-glucose deprivation for example [13].

Following facial nerve axotomy, SHH expression was upregulated in facial motor neurons, promoting the survival of injured motor neurons for a period of 7 days [14].

In a rat model of sciatic nerve crush, *Shh* mRNA was detected at day 1 post-injury in Schwann cells located in the areas proximal and distal to the crush site but not detected in noncrushed nerve. *Shh* mRNA expression was the highest in Schwann cells of the distal segment, where Hedgehog signaling regulated the expression of brain-derived neurotrophic factor (BDNF) following sciatic nerve injury [15].

Furthermore, when SHH is inhibited in intact cavernous nerves, demyelination and axonal degradation occur, suggesting a key role of SHH in the prevention of axonal degradation and prevention of myelin degradation (see Section 3.2.2) [32].

#### 3.1.2. Clearance of Axonal and Myelin Debris

Myelin, particularly myelin-associated glycoprotein and myelin basic protein, inhibits axonal growth and branching [25,33] and thus clearance of myelin debris is a critical step during peripheral nerve healing [29]. Following injury, Schwann cells downregulate promyelinating genes and activate autophagic (myelinophagic) and phagocytic programs to carry out myelin clearance, further facilitated by recruiting blood-borne macrophages [25,33,34]. As axons only grow distally into the injured region when myelin debris are completely degraded [25], it has been suggested that macrophage-mediated myelin clearance is an essential process for nerve regeneration [35].

Although not specifically studied in the context of peripheral nerve injury, SHH could act as a direct macrophage chemoattractant, as shown in other contexts such as the immune response to *Helicobacter pylori* infection in mice [36], tubular cell survival after obstructive kidney injury [37], or tumor-associated angiogenesis in oral cancer [38]. Data from a recent study on inferior alveolar nerve (IAN) regeneration following nerve transection supports such hypothesis [17]. Following IAN transection, myelin-basic protein disappeared from the region of transection at day 3 post-injury in control conditions. When the IAN was treated with cyclopamine (a Hedgehog pathway inhibitor), myelin basic protein was still present in the transected nerve even at day 7 post-injury and a decreased macrophage infiltration was also observed at the same time point [17], suggesting the implication of the Hedgehog pathway in the facilitation of macrophage infiltration.

#### 3.1.3. Formation of Regeneration Tracks

In the case of nerve transection where a gap is created between the two stumps, regenerating axons must cross said gap to allow proper axonal regrowth [39]. Initially, a bridge is formed between proximal and distal stumps, comprised of inflammatory cells and fibroblasts [40]. Early neovascularization and increase in neurotrophic factors such as GDNF within the bridge allow Schwann cell proliferation and migration distal to the injury side [34], collimating with fibroblasts in tubular structures, forming the aforementioned bands of Büngner which serve as tracks for axonal regeneration [5,41].

In a rat model of diabetic neuropathy, systemic injections of SHH induced arteriogenesis in the vasa nervorum of sciatic nerves with concomitant restoration of nerve function, suggesting a key role of SHH in inducing nerve injury-related neovascularization [42], necessary for regeneration tracks formation.

Recently, Bobarnac Dogaru et al. have shown that Hedgehog responsive-fibroblasts (i.e., expressing Gli-1) exist in uninjured facial nerves, that proliferate in response to injury to form regeneration tracks and eventually restore the nerve’s three-dimensional architecture following nerve transection [39].

#### 3.1.4. Neurotrophic Effect

The ability of morphogenetic pathways to promote the regeneration of neural and/or glial components following traumatic and/or demyelinating injury to the brain and/or spinal cord has been under scrutiny for several decades, contesting the long-held belief that the central nervous system lacks such regeneration potential [9,12,13]. For instance, it has been shown that SHH is produced in reactive astrocytes following cortical injury where it regulates neuronal proliferation and differentiation but also that SHH could play a key role in adult neurogenesis in several pathophysiological contexts including activity-dependent neuroplasticity and depression [13].

In the periphery, numerous studies have suggested that SHH could have a neuroprotective effect, in various preclinical peripheral nerve injury models, such as facial nerve axotomy [14], cavernous nerve injury [32,43,44], sciatic nerve transection [16], or sciatic nerve crush [15] but also in diabetic neuropathy models such as streptozotocin-induced diabetes in rats [42].

It has been suggested that intrinsic levels of SHH are sufficient to facilitate axonal regrowth following peripheral nerve injury [16], possible via the production of brain-derived neurotrophic factor (BDNF) [15,44].

BDNF has been shown to be important for survival, differentiation, and protection of neural cells, with neuroregenerative effects in sciatic and facial nerves [45,46].

In a model of cavernous nerve crush, BDNF protein increased 38% the first day then decreased 62% below basal levels at day 4 and returned to normal levels by day 7 post-injury. SHH treatment increased BDNF 36% in normal cavernous nerves whereas BDNF inhibition decreased BDNF 34% in normal cavernous nerves. In crushed cavernous nerves, BDNF increased 44% in SHH-treated crushed nerves, whereas BNDF inhibition in SHH-treated crushed nerves decreased SHH-induced nerve regeneration. This suggests that part of the mechanisms underlying SHH-mediated promotion of cavernous nerve regeneration involve the upregulation of BDNF [44].

#### 3.1.5. Neurite Formation and Pruning

Several studies investigating the role of SHH and related signaling pathway in peripheral nerve healing have suggested that SHH could contribute to neurite formation following various nerve injuries such as sciatic nerve crush [16,47], sciatic nerve microcrush [48], cavernous nerve crush [43,49], or inferior alveolar nerve transection [17].

In a model of sciatic nerve crush, SHH double-labeled with regrowing axon processes in regenerating axons through the crush zone at 7 days. Interestingly, SHH knockdown (by siRNA) reduced overall neurite outgrowth and branch formation [16]. As SHH labeling is widespread in uninjured DRG neurons, it has been suggested that SHH expression could facilitate regenerative plasticity in adult sensory neurons in an autocrine fashion [16].

In 2018, Dobbs et al. showed that exogenous SHH treatment (by means of a peptide amphiphile nanofiber hydrogel) promotes neurite formation in intact cavernous nerves (namely a 2.1 fold increase in number of sprouts and a 1.8 fold increase in average sprout length compared to control animals) but also in crushed cavernous nerves (namely a 1.5 fold increase in number of sprouts, 1.4 fold longer sprout length, and 1.8 fold wider area of sprouting compared to control animals) [49].

Interestingly, the trophic neurite-promoting effect of SHH seems to decrease with age as shown in studies focusing on nerve injuries in aged rats. For instance, in uninjured rat cavernous nerves, SHH treatment induced a 2.7 fold less increase in number of neurites per mm in aged rats compared to adult rats [50].

Finally, SHH could promote also neurite sprouting by an indirect BDNF-mediated pathway [44].

Apart from promoting neurite formation, SHH (and/or Hedgehog pathway) could participate in the pruning of redundant and/or abnormal sprouts. Indeed, it has been suggested that redundant regenerative sprouts gradually disappear as axons enter the bands of Büngner. Misdirected sprouts are thought to disappear through a pruning process resembling Wallerian degeneration [51]. Abnormal sprouting is also thought to be dependent on the type of nerve injury. For instance, in a crush injury the architecture of the distal nerve stump is relatively undisturbed, with Schwann cells still lined up along the basement membrane of the bands of Büngner [52]. In more disruptive transection injuries, multiple sprouts develop at the end of the proximal stump and launch themselves into the gap to the distal stump. Among those multiple sprouts, mature but duplicate sprouts arising from single parent axons have been evidenced and shown to inadvertently innervate inappropriate territories [52]. Functional Hedgehog signaling could prevent such dysfunctional sprouting as shown in an inferior alveolar nerve transection model, where the reconnection of the transected inferior alveolar nerve was compromised due to abnormal sprouting of axons following cyclopamine application (a Hedgehog pathway inhibitor) [17].

#### 3.1.6. Axonal Guidance

Initial evidence of the putative role of SHH in axonal guidance was shown in the central nervous system, where SHH acts as an axonal chemoattractant during midline axon guidance [10] or guidance of retinal ganglion cells [53].

In a sciatic nerve crush model, it was suggested that *Shh* mRNA, induced in dorsal root ganglia (DRG) neuronal cells bodies, could potentiate neurite outgrowth possibly via DRG neuron chemoattraction [47].

#### 3.1.7. Promotion of Remyelination

In a sciatic nerve crush injury of wild-type and *Dhh*-null mice, both *Dhh* and *Patched-2* mRNA levels gradually increased during the regeneration phase when axon–Schwann cell interactions reestablish (between days 14 and 42 post-injury) in wild-type animals, suggesting a potential role of DHH in peripheral nerve regeneration [54]. Considering the more severe myelin damage found in *Dhh*-null mice as compared to wild-type animals, it is plausible that DHH could promote nerve remyelination following peripheral nerve injury [54] but further confirmation is needed to ascertain such hypothesis. Indeed, in the same experiments no significant difference could be observed regarding the density of total myelinated fibers at day 28 post-injury in both *Dhh*-null mice and wild-type mice, suggesting that if DHH plays a role in nerve remyelination, such role must be played in the early stages of nerve regeneration [54].

Furthermore, considering the promyelinating effect of BDNF [55], SHH could potentially participate in nerve remyelination following injury in a BDNF-dependent manner.

Interestingly, as SHH-mediated signaling is essential for oligodendrocyte development in both gray and white matter, recent evidence suggests that SHH could prevent demyelination in several animal models of demyelinating diseases [13] and thus promote remyelination both in the peripheral and central nervous system.

### 3.2. Hedgehog Signaling and Peripheral Nerve Homeostasis

#### 3.2.1. Hedgehog Morphogens and the Concept of Peripheral Nerve Homeostasis

The behavior and expression of SHH following peripheral nerve injury illustrate the persistence and activity of developmental molecules in adulthood, notably for peripheral nerve regeneration. However, the underlying mechanisms and implicated cell types may differ in the adult as compared to during embryonic development [16].

Considering the persistence of functional Hedgehog pathway effectors in adult nerves and related cells (Schwann cells, endothelial cells, fibroblasts, etc.), it can be posited that such morphogenetic signaling is implicated in peripheral nerve homeostasis, being activated following peripheral nerve injury to restore nerve integrity and physiological function.

Several studies support this claim, as SHH has been shown critical to maintain normal cavernous nerve architecture [32,43], penile morphology [56], taste buds [57], or mesenchymal stem cells in adult mice incisors [58].

In the absence of injury, Hedgehog signaling is attenuated [5,59], at least in part by interactions between SHH and HIP (Hedgehog-interacting protein), an SHH inhibitor that provides a physiological negative regulatory feedback loop [31]. A recent study has shown that DHH is expressed in intact sciatic nerves but not SHH. Patched-1 and Gli-1 were both expressed in the endoneurium and perineurium of such intact nerves. Numerous Gli-3 positive cells could be found in the endoneurium but not the perineurium and Gli-3 was expressed only in myelinated Schwann cells but not in endothelial cells or macrophages, suggesting that Gli-3 probably represses Hedgehog signaling activity in the Schwann cells of healthy nerves [60]. Interestingly, Gli-3 mutation results in ectopic activation of Hedgehog signaling, resulting in morphological changes in intact sciatic nerves, suggesting that suppression of Hedgehog signaling by the repressor form of Gli-3 is essential for normal homeostasis in intact peripheral nerves [60].

In a rat model of cavernous nerve injured-induced erectile dysfunction, it has been suggested that SHH is essential in maintaining neuro-glial interactions and regulating the neural microenvironment under physiological conditions and peripheral nerve injury [49,61]. SHH and its receptors Patched-1 and Smoothened were localized in both pelvic ganglia neurons and glia under physiological conditions [43,61] and Smoothened underwent an anterograde transport to more peripheral nervous structures [61]. Such transport was halted following cavernous nerve crush injury and resumed following SHH treatment [61]. Interestingly, a similar role of SHH in the maintenance of neuro-glial interactions has been shown in the central nervous system, where anterograde axonal transport of SHH could allow delivery of SHH protein to the subventricular zone (a key region of adult neurogenesis) and the striatum, thus promoting the maintenance of the nigrostriatal circuit during adulthood [9].

Hedgehog signaling could also regulate peripheral neuro-vascular interactions. It has been hypothesized that SHH produced in Schwann cells of penile nerves regulates the homeostasis of adjacent vascular structures through the induction of VEGF-A. In a model of cavernous nerve injury, SHH treatment at time of cavernous nerve injury was able to prevent/postpone cavernous nerve injury-induced apoptosis of smooth muscle and endothelial cells in the corpora cavernosa [56].

Considering the differential expression of DHH and SHH in healthy and injured nerves, it is most probable that DHH and SHH have distinct functions in peripheral nerves [60,62]. This should be considered when investigating the role of Hedgehog signaling in various physiological and pathological processes in future studies.

#### 3.2.2. Prevention of Myelin Degradation

The Hedgehog signaling pathway could participate in the prevention of myelin degradation, even in physiological conditions. For instance, treatment with a SHH inhibitor leads to demyelination in intact cavernous nerves [32]. Furthermore, lack of *Dhh* has been shown to accelerate the degradation of myelin (myelinophagy and phagocytosis) during early nerve regeneration [54,59,63]. Finally, in an in vitro model of compression-induced demyelination, DHH treatment had a preventive effect against myelin loss [59].

#### 3.2.3. Homeostasis of the Blood–Nerve Barrier

The blood–nerve barrier, the monolayer of endoneurial endothelial cells linked together by tight- and adherens-junction proteins, protects the peripheral nerve parenchyma from blood-borne immunological, infectious or toxic insults [64] (as similarly evidenced in the seminal work of Alvarez et al. on blood–brain barrier homeostasis [65]), making it the second most-restrictive barrier of the body after the blood–brain barrier [66].

Nevertheless, a transient opening of the blood–nerve barrier may participate in the healing process following peripheral nerve injury [67]. Indeed, following axonal disintegration in the distal nerve stump, the blood–nerve barrier is permeabilized allowing the recruitment of blood-borne macrophages responsible for the phagocytosis of myelin debris within days of the injury, before exiting the nerve by the circulation once remyelination has occurred [67]. Such transient increase in blood–nerve barrier permeability has been recently shown to be under the control of reversible inhibition of endoneurial endothelial Hedgehog signaling that controls endoneurial/perineurial endothelial tight-junction protein expression (including Claudin-1, Claudin-5, and Occludin) by Wnt/β-catenin signaling [20] or by activation of innate immunity TLR4 pathway [18], possibly by myelin basic protein released from injured Schwann cells [67,68].

Studies of injured nerves in *Dhh* −/− mice have revealed structural abnormalities of the perineurium and blood–nerve barrier, resulting in the entry of proteins and macrophages into the nerve parenchyma [11,54,63], supporting the role of the Hedgehog pathway in blood–nerve barrier homeostasis.

It should be noted that the transient but efficient inflammatory response in the peripheral nervous system required for nerve healing is in stark contrast with that of the central nervous system that is associated with inhibitory scar formation and poor nerve regeneration [67]. Understanding the underlying mechanisms responsible for such differences could provide interesting perspectives for innovative regeneration strategies of both central and peripheral nervous lesions.

### 3.3. Hedgehog Signaling in Peripheral Nerve Healing and Homeostasis

As previously detailed, current data suggests that Hedgehog signaling is implicated in numerous processes relating to physiological peripheral nerve healing following nerve injury but also in maintaining peripheral nerve homeostasis in physiological conditions.

The multifaceted role of Hedgehog signaling during peripheral nerve healing and peripheral nerve homeostasis is summarized in Figure 1.

### 3.4. Localization and Function of Hedgehog Pathway Effectors in Peripheral Nerve Healing

As peripheral nerve healing involves multicellular processes, implicated in neuro-glial and neuro-vascular interactions, components of the Hedgehog signaling pathway have been evidenced in numerous nerve injury models, in physiological and pathological conditions.

Unfortunately, owing to the great variability in animal models (rats, mice, etc.), type of injured nerve (sciatic nerve, cavernous nerve, inferior alveolar nerve, facial nerve, etc.), type of injury (transection, crush injury, ligation, etc.), and type of effector studied (gene, mRNA, protein, etc.), results must be analyzed with caution and reserve, as they might not be transferable to other models (see Section 4.1).

The data regarding localization and function of Hedgehog pathway effectors in peripheral nerve healing, collected in the present review, is summarized in Table 1.

## 4. Nerve Healing over the *hedge*: Hedgehog Pathway Disruption and Neuropathic Pain Installation

Although the peripheral nervous system does indeed show remarkable plasticity and regeneration capabilities, the regeneration process following peripheral nerve injury is not always complete and abnormal nerve healing can lead to severe and irreversible disability [17,70,71,72] and neuropathic pain development.

Understanding pathological nerve healing following peripheral nerve injury and its relationship with the apparition of neuropathic pain could help develop new regeneration strategies but also specific antalgic drugs aimed at suppressing peripheral nerve injury induced neuropathic pain.

### 4.1. Exploring Hedgehog Signaling Modifications in Different Peripheral Nerve Injury Models

As previously stated, Hedgehog signaling has been studied in numerous peripheral nerve injury models, including nerve crush, transection, ligation, or constriction injuries. Interestingly, expression of SHH/DHH or Hedgehog pathway effectors varies in the different injury models [17] that could in part explain differences in intrinsic healing potential of such injuries (see below).

In compression neuropathies, Schwann cells proliferate in response to injury and downregulate myelin proteins leading to demyelination followed by subsequent remyelination of the axon as well as axonal sprouting [59]. Such early changes occur in the absence of morphological and electrophysiological evidence of axonal damage [73]. Contrastingly, acute nerve injuries such as transection or crush injuries are characterized by axonal injury followed by ensuing Wallerian degeneration. Alternatively, compression neuropathies are characterized by a local demyelination in the absence of Wallerian degeneration [74].

#### 4.1.1. Nerve Transection

Nerve transection models have investigated the repair of a nerve after a localized cut (usually made with surgical scissors) to various nerves such as the facial nerve, the inferior alveolar nerve, or the sciatic nerve.

In a model of facial nerve axotomy, ipsilateral facial nucleus motor neurons started to show the expression of *Shh* transcripts at 6 h post-injury, with a peak at 24 h and sustained levels of expression until 4 weeks post-injury [14]. *Smoothened* receptor mRNA was upregulated at 24 h post-axotomy, but not *Patched-1* [14]. Interestingly, it was found that motor neurons of adult facial nucleus expressed *Shh* mRNA at a low level in uninjured nerves [14], which is not the case in other nerves or injury models (see below).

In another study, using a sciatic nerve transection model, SHH labeling was widespread in dorsal root ganglia (DRG) neurons, especially in large caliber neurons under physiological conditions. At 7 days post-injury, SHH double-labeled with regrowing axonal processes [16].

In a model of inferior alveolar nerve (IAN) transection, no Gli-1 or SHH could be detected in the IAN before transection. At day 3 post-injury, the proximal stump showed strong SHH expression, while the entire distal region of the IAN including the distal stump exhibited weak SHH immunoreactivity. Weak expression of SHH was also observed in the distal transected IAN, but not detected in the proximal region of the transected IAN (Figure 2a). Gli-1 expression patterns were similar to those of SHH at day 3 post-injury, although more intense (Figure 2a). At day 7 post-injury, both SHH and Gli-1 expression was reduced in all regions of the transected IAN [17]. Considering the differential expression pattern of SHH signaling observed between the proximal and distal regions of the IAN, including the stumps of the transected nerve, it is possible that SHH signaling plays distinct roles in these regions, namely the promotion of axonal growth at the proximal stump of the transection site, and regulation of myelin degradation in the distal region [17].

A recent study performed using a sciatic nerve transected model investigated the molecular interactions between Gli-1 and Gli-3 during peripheral nerve healing [60]. Gli-1 expression was increased in injured sciatic nerves during the first week following injury. Gli-3 was significantly decreased in the damaged nerves at day 1 post-injury. Western blot analysis confirmed a reduction in the repressor form of Gli-3 from day 1 to day 7 post-injury. Nevertheless, Gli-3 expression was retained in the proximal region of the injured nerve. Conversely, Hedgehog signaling was activated (in Schwann cells and other (unspecified) cell types) in the proximal stump and distal region of injured nerves, but not in the proximal region [60].

Ectopic SHH expression upon nerve injury began at day 1 and persisted for 7 days, whereas DHH expression was reduced at day 1, persisting for 7 days (as in other studies showing *Dhh* downregulation in injured peripheral nerves [54,59]). Hedgehog pathway ligand was thus rapidly switched from DHH to SHH in the first week of healing, which occurred in the proximal stump and distal region of injured sciatic nerves [60]. SHH expression was reduced at 14 days after injury, but DHH expression began to increase at 14 days after injury [60].

Counteracting native DHH reduction post-injury with DHH protein resulted in nerve regeneration failure, suggesting that reduction in DHH is essential for peripheral nerve regeneration. Interestingly, initiation of SHH expression after injury occurred when DHH expression was reduced. Inhibition of SHH function by anti-SHH antibody led to nerve regeneration failure, suggesting that upregulation of SHH expression following injury is essential in early stages of nerve regeneration. The authors thus conclude that ligand switching between SHH and DHH is critical for peripheral nerve regeneration, as is the increase in repressor Gli-3 observed at later stages of nerve regeneration [60].

Finally, several authors recently investigated the effect of chronic axotomy (transection) of the facial nerve in rats (at 12, 20, 28, and 36 weeks post-initial injury), as compared to single axotomy [75]. SHH immunolabeling was observed in motoneurons of the facial nucleus more importantly in the control side that in the injured side (single axotomy). In the reinjured group (chronic axotomy), SHH expression in the reinjured side was higher at 12 weeks compared to the control side, but at the 36 weeks reinjury no differences could be found between injured and control sides. After 36 weeks, SHH expression had decreased even more, suggesting a regeneration potential that peaked within 5 months of initial injury [75]. Unfortunately, lack of rigorous quantification in this study hampered further analyses of SHH expression.

#### 4.1.2. Crush Injury

Crush injury models are often used to mimic iatrogenic surgical lesions or compressive neuropathies in humans. Simple crush injuries are performed using forceps of various widths depending on the intended crush length. Quite paradoxically, longer crush zones are more prone to regenerative sprouting than smaller crush zones, possibly linked to an increase in *Shh* mRNA expression in longer crush injury (of unknown underlying mechanism) [52].

Following sciatic nerve crush in rats, *Shh* mRNA was detected 1 day post-injury in the areas proximal and distal to the crush site (Figure 2b) but not detected in noncrushed sciatic nerves. *Shh*, *Gli-1*, *Bdnf*, and *Gdnf* mRNAs upregulated after injury, more importantly in the distal segment (regardless of timing post-injury). The observed mRNAs colocalized with Schwann cells, suggesting that Schwann cells (especially those of the distal segment) were the major cells producing SHH following sciatic nerve injury. Inhibition of SHH signaling by continuous administration of cyclopamine decreased the mRNA upregulation of *Gli-1* and *Bdnf* after injury but not that of *Shh* and *Gdnf*. This suggests that SHH signaling regulated Gli-1 and BDNF expression following sciatic nerve injury [15].

In a model of cavernous nerve crush (mimicking an iatrogenic nerve lesion frequently occurring following nerve-sparring prostatectomy in humans), SHH protein was found abundant in cells on either side of the crush site [32]. In another study investigating the spontaneous recovery following a cavernous nerve crush injury, a significant increase in the hypoxia marker *Hif-1α* and a decrease in *Shh* mRNA were observed following injury compared to sham animals at week 12 post-injury. At week 24 post-injury, the increase/decrease were both diminished, and at 6 months erectile function was restored [76].

#### 4.1.3. Comparing the Regeneration Response in Acute Injury Models

Both transection and crush injuries are acute injuries that induce an immediate regenerative response, allowing eventual complete nerve regeneration and functional recovery in such animal models (although transection injuries require more significant nerve regeneration because of the complete loss of nerve integrity).

As previously stated, the role of Hedgehog signaling could differ between injury models. In nerve transections, high expression of SHH and its effectors in the distal aspect of the proximal stump (Figure 2a) could promote axonal regrowth, whereas its high expression in the distal region of the distal stump (Figure 2a) could promote myelin degradation as part of the Wallerian degeneration process that occurs following nerve transection [17]. Conversely, in a nerve crush, expression of SHH (and its effectors) is found in all regions of the nerve (proximal, crush site, and distal) [15] (Figure 2b), that seem to establish a concentration gradient, prototypical of such morphogens. Such gradient (if actually present) could participate in guiding axonal regrowth, following SHH production from uninjured adjacent neurons and/or Schwann cells, and thus account for the better prognosis of equivalent lesions in humans.

#### 4.1.4. Chronic Nerve Injuries: Compressions and Constrictions

Chronic injuries to peripheral nerves have also been modeled to mimic compression neuropathies of various origins (entrapment, tumoral, iatrogenic, etc.) in humans. Compressive and constrictive injuries are obtained by encasing nerves with rigid tubes or by ligating the target nerve.

As chronic injury models provide sustained nerve trauma, they have often been developed as painful neuropathy models such as the chronic constriction injury [77,78]. Unfortunately, few studies using chronic nerve injury models have focused on Hedgehog pathway-mediated nerve regeneration.

In an acquired chronic compressive injury of the sciatic nerve, *dhh* is downregulated at the onset of demyelination process. *Dhh* deletion in vivo resulted in important demyelination and damage, as evidenced by both rapidity of onset and magnitude of disease after a demyelinating insult as compared to their wild-type counterparts [59].

In a study investigating the role of Hedgehog signaling in the disruption of the blood–nerve barrier observed following chronic constriction injury of the sciatic nerve in rats, a transient upregulation of *Shh* mRNA was observed between 3 and 24 h post-injury in constricted sciatic nerves that completely disappeared after 48 h. In parallel, both *Patched-1* and *Gli-1* mRNA expression were profoundly downregulated between 3 h and 30 days (60 days for *Gli-1*) in the injured sciatic nerves. Colocalization experiments suggested that at least *Patched-1* and *Gli-1* mRNA downregulations were associated with a decrease in Patched-1 and Gli-1 protein production within endoneurial endothelial cells, that played a key role in the concurrent decreased production of tight junction proteins (such as Claudin-5 and Occludin), as such proteins are under the transcriptional control of Gli-1 [18].

Similar results were obtained in another study from the same group, investigating the role of Hedgehog signaling in the disruption of the blood–nerve barrier in a trigeminal injury model: the infra-orbital nerve chronic constriction injury [19]. In this experimental paradigm, both *Gli-1* and *Patched-1* mRNAs were downregulated between 3 and 48 h post-injury (24 h for Patched-1) [19].

Interestingly, perineural injections of cyclopamine in the vicinity of the sciatic nerve [18] or the infra-orbital nerve [19] in naïve rats could mimic the molecular, vascular, and behavioral alterations observed following respective chronic constriction injuries, suggesting a major role of Hedgehog signaling pathway in the development of painful post-traumatic peripheral neuropathies (see below).

#### 4.1.5. Summarizing Hedgehog Signaling Modifications in Peripheral Nerve Injury Models

Investigation of the relevant scientific literature has revealed numerous modifications of Hedgehog signaling in various peripheral nerve injury models, depending on the type of injured nerve and type of injury.

The main modifications in Hedgehog signaling observed in peripheral nerve injury models are summarized in Table 2.

### 4.2. The Critical Issue of Timing in Nerve Regeneration

Although seldom studied in the scientific context of Hedgehog-mediated nerve regeneration, there is some evidence suggesting that timing could be a critical issue for nerve regeneration, with obvious clinical implications.

In a recent study on cavernous nerve regeneration following crush injury, the authors showed that SHH intervention by means of a peptide amphiphile nanofiber gel had the most regenerative effect when given immediately after cavernous nerve crush injury [49].

Similarly, in a model of cavernous nerve crush, SHH pathway inhibition (by 5E1 SHH inhibitor) decreased neurite formation only during the first 2 days following nerve injury, suggesting that the first 48 hours after crush injury are a critical window when trophic factors such as SHH are released to promote neurite formation [50].

### 4.3. Dysfunctional Hedgehog Signaling as a New Culprit of Post-Traumatic Peripheral Neuropathic pain Development

Most studies have investigated the role of functional Hedgehog signaling during peripheral nerve healing following various nerve injuries. Nevertheless, dysfunctional Hedgehog signaling has been implicated in numerous diseases [13] and could participate in the development of post-traumatic peripheral neuropathic pain [18,19,20].

Despite an important capacity for regeneration, peripheral nerve injuries do not always heal uneventfully in clinical practice [80]. It has been suggested that the disturbance of the regeneration process could lead to abnormal neural function [17], possibly resulting in neuropathic pain. Anecdotal evidence supports this hypothesis, showing a decrease in SHH protein expression in the thalamus of rats subjected to sciatic nerve chronic constriction injury and the implication of Hedgehog signaling in both the thalamus and ipsilateral spinal dorsal horn that contributes to neuropathic pain following peripheral nerve injury [79].

#### 4.3.1. Understanding Dysfunctional Nerve Regeneration

One of the main reasons for insufficient nerve regeneration in nerve transections could be that axons must regenerate over a relatively long distance and the most distal Schwann cells gradually lose their ability to foster nerve regeneration [81,82], which could be in part due to reduced expression of neurotrophic factors such as GDNF and BNDF [83]. Two factors could contribute to this deterioration: gradual death of chronically denervated Schwann cells [84] and reduction in expression of growth-supportive factors such as BDNF and GDNF by surviving cells. It was shown recently that STAT-3, which does not have any major function during Schwann cell development, plays an important role in promoting the survival of chronically denervated repair Schwann cells [81]. STAT-3 could thus be the second transcription factor (after c-Jun) to be specifically activated in repair Schwann cells following peripheral nerve injury and a conditional knock-out of STAT-3 results in a decrease in key markers of repair Schwann cells such as *Shh* [81].

Age could be another factor implicated in decreased nerve regeneration potential. It was shown that SHH protein decreased with age (in rats): in aged rats a 77% decrease in SHH was observed compared to young rats [43]. Furthermore, aging has been shown to slow myelin clearance and the onset of functional recovery after injury [85] and such delay seems primarily due to a deteriorating response of transcriptional mechanisms of Schwann cells [4,86].

#### 4.3.2. Abnormal Sprouting and Neuroma Formation

Traumatic neuromas, non-neoplastic exuberant haphazard proliferations of nerve fascicles (axons and Schwann cells), can sometimes occur in response to nerve injury [87]. Interestingly, in an inferior alveolar nerve (IAN) transection model, reconnection of the transected IAN treated with cyclopamine (a Hedgehog pathway inhibitor) was compromised due to abnormal sprouting of axons, suggesting that reduction in Hedgehog signaling could participate in traumatic neuroma formation [17]. Such neuromas have been long shown to be a source of ectopic painful discharges [88].

This could suggest a critical role of Hedgehog signaling in the genesis of such painful neuromas that affect 2–60% of patients with a nerve injury [89].

#### 4.3.3. Neuropathic Pain Development Following Post-Traumatic Blood–Nerve Barrier Disruption

Recent studies have shown the role of blood–nerve barrier (BNB) disruption in the formation of sustained local neuroinflammation, peripheral sensitization, and neuropathic pain development [18,19,20,90,91,92]. Endoneurial endothelial Hedgehog signaling was shown to play an essential role in such BNB disruption and neuropathic pain development. Hedgehog pathway markers (Patched-1, Gli-1) were significantly downregulated following chronic constriction injury (CCI) of the sciatic nerve [18] or the infra-orbital nerve [19,20] and inhibition of the Hedgehog pathway by perineural injection of cyclopamine in the vicinity of the sciatic nerve of healthy rats recapitulated the molecular (endothelial tight-junction proteins downregulation, Hedgehog pathway downregulation, and upregulation of markers of neuroinflammation), vascular (increased epi and endoneurial vascular permeability), and behavioral changes (sustained mechanical allodynia of the hindpaw) observed in the CCI model [18]. This suggests a critical role of Hedgehog pathway in neuropathic pain development [18,19,20]. Interestingly, pretreatment or post-treatment with a SHH agonist (SAG, Smo Agonist) did not prevent or mitigate the neuropathic phenotype following CCI in the sciatic or infra-orbital nerves [18,19].

#### 4.3.4. Could SHH Be a Treatment for Neuropathic Pain?

Several studies have suggested that treatment with SHH could attenuate neuropathic pain in various preclinical models [93].

In a recent study, activation of Hedgehog pathway (via intraperitoneal injections of purmorphamine, a Smoothened agonist) decreased the mechanical allodynia following chronic constriction injury [79]. Interestingly, as previously stated, perineural injections of another Smoothened agonist (SAG) could not decrease the mechanical allodynia following chronic constriction injury in another study [18], suggesting that the anti-allodynic effect of Hedgehog pathway activation is probably not peripheral.

In a model of diabetic neuropathy, a decrease in *Dhh* mRNA was observed in the sciatic nerve of diabetic rats, maintained for 10 weeks [69]. Motor and sensory-nerve conduction velocity of treated diabetic rats, decreased in the diabetic neuropathy model, were restored to control values over a 5-week treatment period with SHH (SHH-IgG), with maintenance of the axonal caliber of large myelinated fibers [69].

Furthermore, using the same model of diabetic neuropathy, SHH treatment (by subcutaneous injections) was shown to restore the vasa nervorum (epineurial, perineurial, and endoneurial capillaries) in the nerves of such diabetic rats [42]. More accurately, SHH treatment induced a neovasculature, distinct from the native one, as epineurial and perineurial capillaries were larger in diameter and contained more α-actin positive cells [42]. Moreover, SHH treatment also increased motor nerve conduction velocity and sensory nerve conduction velocity [42].

## 5. Future Research and Clinical Perspectives

### 5.1. Could Hyperactive Nerve Regeneration Drive Neuropathic Pain Development?

Recent evidence has raised the possibility that neuropathic pain could indeed result from dysfunctional nerve healing, but actually from a hyperactive nerve regeneration that fails to properly reinnervate the injured nerve [94]. Following spinal nerve ligation (ligation of L5 spinal nerve and cut at 1 mm distal to the suture), a common temporarily painful neuropathy model, the nerve could regenerate into the sciatic nerve (with restoration of electrical conduction, mechanical responses, and proper tracer migration) and the regenerating nerve was in fact the source of pain in this model [94]. Interestingly, disrupting the nerve regeneration inhibited the pain. Comparatively, in a spared nerve injury model (where pain is thought to be permanent), the regeneration process resulted in tangled neuromas at the injury site without proper reinnervation (as observed in the spinal nerve ligation model). Blocking the regeneration process via perfusions of Semaphorin 3A (an inhibitory axonal guidance molecule) prevented or reversed the painful behavior (even without removing the tangled fibers). It is thus concluded that in this model, the long-lasting pain behavior could stem from the anatomical inability of regenerating nerves to successfully reinnervate their target tissues, resulting in a persistent, yet futile, regeneration process [94].

A recent clinical study supports this hypothesis. In a study on patients suffering from chronic inflammatory demyelinating polyneuropathy (CIDP), *shh* mRNA expression was significantly higher in skin biopsies of CIDP patients that those of control subjects (with similar intra-epithelial nerve fiber density), and this high expression of *shh* mRNA was decreased following treatment of CIDP [95]. This suggests that in the context of this immune-mediated neuropathy, the small depleted axons attempt to mount a regenerative response [95]. Whether this regenerative response takes part in the underlying pathophysiology of CIDP still remains to be answered.

### 5.2. Understanding Hedgehog Signaling Inhibition in the Chronic Constriction Injury (CCI) Model

Considering the essential role of Hedgehog signaling in molecular, vascular, and behavioral changes observed in CCI models [18,19,20], understanding the underlying mechanisms of such inhibition could be of interest and help explain the lack of pre/post-treatment effect of Hedgehog pathway activation on neuropathic pain in such experimental conditions [18,19].

It is well known that Hedgehog signaling requires a functional ciliary architecture [96,97] and that loss of primary cilia is associated with Hedgehog pathway inhibition [98,99]. Furthermore, it has been shown that the Smoothened receptor is necessary for the development and maintenance of ciliary architecture [100]. Interestingly, chronic constriction injury of the sciatic nerve in rats leads to a significant and persistent downregulation of *Smoothened* mRNA in the sciatic nerve parenchyma from 6 h to 15 days post-injury (Figure 3), that could participate in the loss of primary ciliary architecture and subsequent Hedgehog pathway inhibition.

Furthermore, alterations in endoneurial vessel morphology have been serendipitously observed both in infra-orbital nerves of rats subjected to infra-orbital nerve chronic constriction injury and in uninjured infra-orbital nerves treated with cyclopamine (a Smoothened antagonist known to decrease production of both *Smoothened* mRNA and protein [101]) (Figure 4), possibly linked to a loss of Smoothened receptor [100].

Exploring the microvascular changes related to chronic constriction injury could thus be of interest to better understand the role of loss of Hedgehog signaling in neuropathic pain development.

### 5.3. Investigating and Developing New Nerve Regeneration Strategies

Although adult peripheral nerves have an intrinsic ability to regenerate, the endogenous response is often limited and does not allow for full recovery of function. Manipulation of the nerve microenvironment to promote neuronal survival and repair could be key to improve regeneration strategies [44].

For instance, a few studies have invested the role of brief extracellular electrical stimulation in fostering nerve regeneration, as it has been shown that such electrical stimulation applied immediately after nerve injury could increase nerve regeneration in animal models and humans [102]. In a diabetic neuropathy model, an enhanced expression of *Shh* mRNA was observed in ipsilateral DRGs following electrical stimulation, suggestion a potential role in electrical stimulation-mediated peripheral nerve regeneration. Interestingly, no rise in SHH protein was observed in the sciatic nerve [72], which suggests that *Shh* mRNA could function without protein transcription as seen in other experimental paradigms [8].

Finally, several studies have investigated new means of SHH protein delivery in a model of cavernous nerve crush injury, using a peptide amphiphile nanofiber hydrogel that allowed efficient local delivery of SHH [32,43,44,49,50,61]. Such delivery strategy could be applied to other models for future investigations.

## 6. Conclusions

The Hedgehog pathway that plays key morphogenetic roles during development also assumes critical roles in peripheral nerve healing following injury and in peripheral nerve homeostasis, such as preventing myelin degradation or maintaining blood–nerve barrier impermeability. Schwann cell, neuronal and endothelial-mediated Hedgehog signaling cooperate to promote physiological Wallerian degeneration and proper nerve regeneration.

Elucidating the numerous roles of Hedgehog signaling during normal and dysfunctional nerve healing could bring a better understanding of nerve regeneration processes and their still-ambiguous relation to neuropathic pain development.

As hyperactive regeneration could participate in neuropathic pain development, nerve regeneration can be construed as a physiological process that requires precise temporal and spatial controls, and thus morphogens such as Sonic Hedgehog or Desert Hedgehog would be ideal regulators of such peripheral nerve homeostasis.

## Figures and Tables

**Figure 1 ijms-21-09115-f001:**
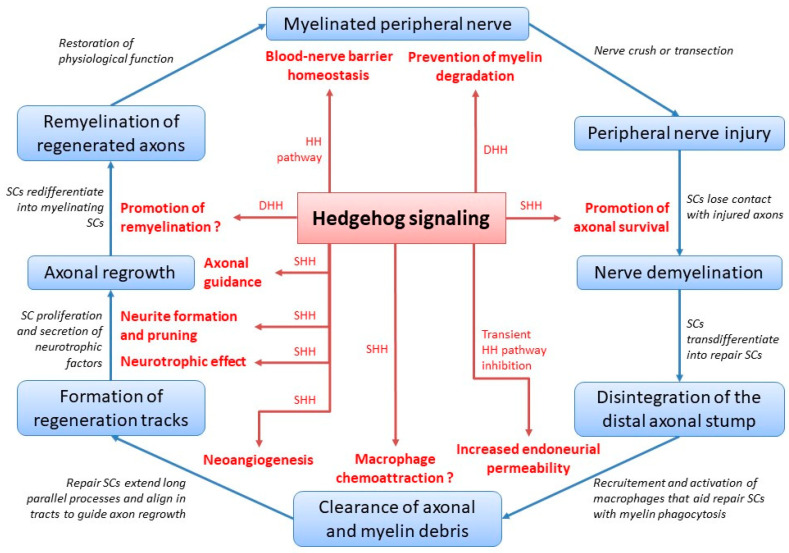
Hedgehog signaling during peripheral nerve healing and homeostasis. HH = Hedgehog; DHH = Desert Hedgehog; SC = Schwann cell; SHH = Sonic Hedgehog; putative roles are featured with a question mark.

**Figure 2 ijms-21-09115-f002:**
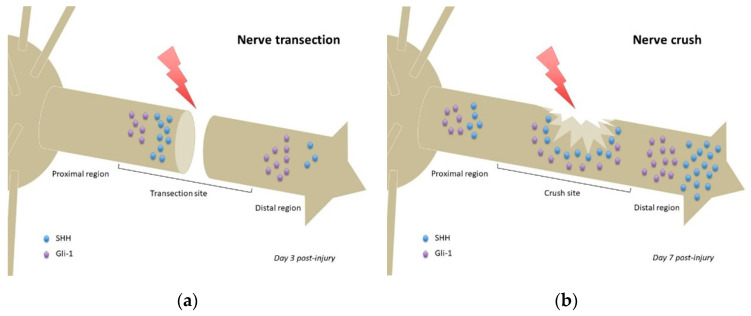
Comparison of Hedgehog signaling and Sonic Hedgehog localizations following nerve transection (**a**) or nerve crush (**b**). Relative quantities of Sonic Hedgehog and Gli-1 protein/mRNA are represented at the time of highest expression in each nerve injury model (based on data from Yamada et al. 2018 [17] and Hashimoto et al. 2008 [15], respectively). Noteworthy: the SHH gradient in the nerve crush model (**b**) and lack of said gradient in the transection model (**a**), which has more focalized accumulations of SHH protein. Gli-1 = glioma-associated oncogene homolog-1; SHH = Sonic Hedgehog.

**Figure 3 ijms-21-09115-f003:**
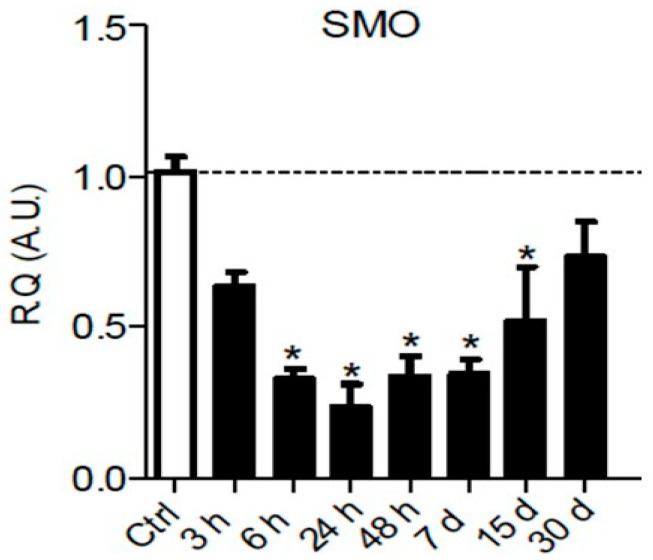
Chronic constriction injury (CCI) of the sciatic nerve induces early and prolonged downregulation of Smoothened receptor mRNA. Changes over time of Smoothened (SMO) mRNA levels were assessed in the sciatic nerve of sham or CCI-injured rats using semiquantitative reverse transcription polymerase chain reaction analyses. Data are presented as relative quantification (R.Q.) in arbitrary units (A.U.) corresponding to the ratio of specific mRNA over RPS18 mRNA. Each bar corresponds to the mean ±SEM of n = 6–8 animals for each time post-injury; * *p* < 0.05. One-way analysis of variance followed by Bonferroni post hoc test was used.

**Figure 4 ijms-21-09115-f004:**
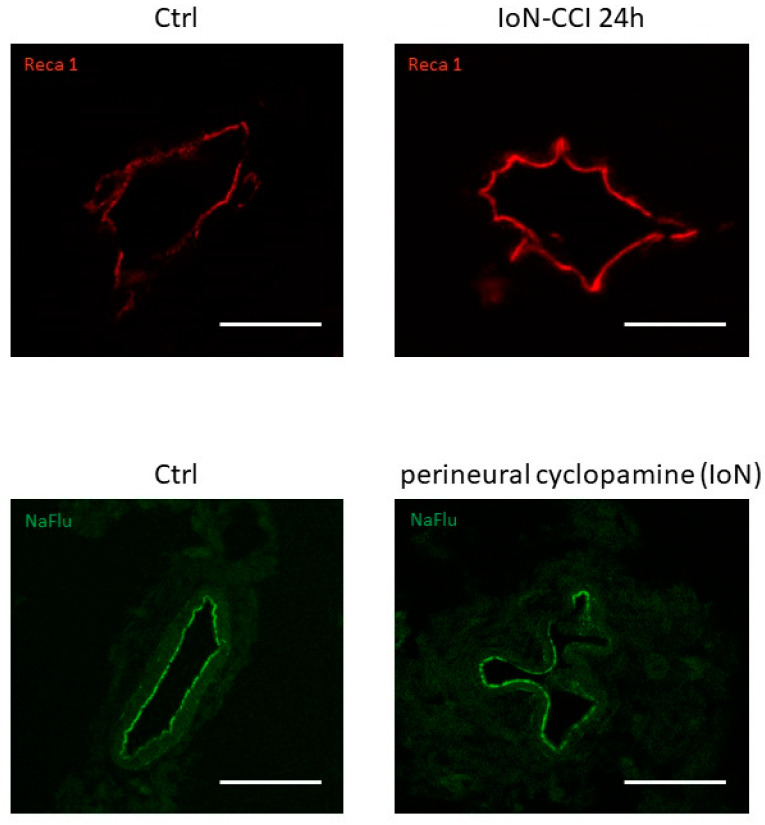
Vascular morphological changes observed using confocal microscopy in axial slices of infra-orbital nerves (IoN) subjected to chronic constriction injury (IoN-CCI) at 24 h post-injury (**upper left panel**) or sham surgery (**upper right panel**) or perineural injections of cyclopamine at 6 h post-injection (**lower left panel**) or vehicle (**lower right panel**) using Reca-1 and sodium fluorescein (NaFlu) for vessel immunolabeling, respectively. Chronic constriction injury and perineural injections were performed according to previously described methodologies (see Moreau et al. 2016 [18], 2017 [19] for methodological details).

**Table 1 ijms-21-09115-t001:** Localization and functions of Hedgehog pathway effectors in peripheral nerve healing. (DRG = dorsal root ganglia, NOS = not otherwise specified ^1^, PG = pelvic ganglia, PNI = peripheral nerve injury, TG = trigeminal ganglia).

Hedgehog Pathway Effectors		Localization	Functions	References
Hedgehog proteins	SHH	Schwann cells (injured nerve)	Nerve regeneration following PNI	[6,32,60]
		Injured nerve	Nerve regeneration following PNI Promotion of neuronal survival	[14,60]
		Neuronal cell bodies (DRG, TG, PG)	Promotion of neurite outgrowth Neuroprotection	[16,43,47,58]
		Regrowing axons	Nerve regeneration following PNI	[16,17]
		Glial cells (PG)	Neuro-glial interactions following PNI	[43]
	DHH	Schwann cells	Blood–nerve barrier homeostasis	[11,54,63]
		Healthy (sciatic) nerve (NOS)	Blood–nerve barrier homeostasis; prevention of myelin degradation	[54,59,60,69]
		Crushed sciatic nerve	Nerve regeneration following PNI	[59]
Transmembrane receptors	Patched-1	Endoneurium and perineurium of healthy (sciatic) nerve	Peripheral nerve homeostasis	[60]
		Schwann cells (healthy nerve)	Peripheral nerve homeostasis	[60]
		Schwann cells (injured nerve)	Nerve regeneration following PNI	[60]
		Glial cells (PG)	Neuro-glial interactions following PNI	[43]
		Neurons (PG)	Neurite outgrowth	[50]
		Endoneurial endothelial cells (sciatic nerve)	Blood–nerve barrier homeostasis	[18]
	Patched-2	Healthy sciatic nerve Schwann cells	DHH-mediated signaling	[54]
		Injured sciatic nerve Schwann cells	DHH-mediated signaling	[54]
		Nerve-derived fibroblasts	DHH-mediated signaling	[54]
	Smoothened	Neurons (PG)	Neurite outgrowth	[43,50]
		Neurons (facial motor nerve)	Promotion of neuronal survival	[14]
		Sciatic nerve (NOS)	Physiological hedgehog signaling (including blood–nerve barrier homeostasis)	[54] and present article
Transcription factors	Gli-1	Perineurium (strong signal) and endoneurium (weak signal) of healthy (sciatic) nerve	Peripheral nerve homeostasis	[60]
		Injured (sciatic) nerve	Nerve regeneration following PNI	[60]
		Endoneurial endothelial cells (sciatic nerve)	Blood–nerve barrier homeostasis	[18]
		Endoneurial fibroblasts (facial nerve)	Nerve regeneration following PNI	[39]
	Gli-3	Schwann cells (healthy nerve)	Hedgehog signaling repression under physiological conditions	[60]
		Proximal region of injured nerve	Hedgehog signaling repression during peripheral nerve healing	[60]

^1^ This mention states that the precise cell type was not specified and/or investigated in the study.

**Table 2 ijms-21-09115-t002:** Hedgehog signaling modifications in peripheral nerve injury models (NOS = not otherwise specified ^1^).

Peripheral Nerve Injury Model	Localization	Modifications of Hedgehog Signaling	References
Sciatic nerve crush	Schwann cells	Upregulation of *Shh* mRNA	[15]
		Upregulation of *Shh* mRNA 6 h post-injury Upregulation of Gli-1 and Patched-1 24–72 h post-injury	[31]
	Neuronal cells bodies	Induced *Shh* mRNA at day 3 post-injury	[47]
	Sciatic nerve (NOS)	*Shh* mRNA detected at day 1 post-injury in areas proximal and distal to the crush but not detected in noncrushed sciatic nerve *Gli-1* mRNA upregulated after injury, more importantly in the distal segment	[15]
	Axonal end bulbs	Increased production of SHH at day 7 post-injury Important increase in *Shh* mRNA at 24 h post-injury	[52]
Sciatic nerve transection	Proximal stump and distal region of injured nerve (NOS)	Increased expression of SHH from day 1 to day 7 and decreased expression of DHH from day 1 to day 7 (=ligand switching) Reduced expression of SHH at 14 days and increased expression of DHH at 14 days Decreased Gli-3 from day 1 to day 7	[60]
	Regrowing axons	SHH labeling at 7 days in regrowing axons	[16]
	DRG neurons	SHH labeling	[16]
Cavernous nerve crush	Schwann cells	SHH protein localization on either side of crush	[32]
	Cavernous nerve (NOS)	SHH protein decrease during the first week (in aged rats)	[43]
Inferior alveolar nerve transection	Inferior alveolar nerve (NOS)	Strong SHH/Gli1 expression at day 3 post-injury in the proximal stump Weak expression in distal transected IAN but not the proximal region of transected IAN Both signals reduced at day 7 post-injury	[17]
Facial nerve axotomy	Facial nucleus motoneurons	*Shh* transcripts increase at 6 h post-injury with a peak at 24 h, maintained for 4 weeks *Smoothened* receptor mRNA upregulated at 24 h post-axotomy (but not *Patched-1*)	[14]
		SHH increase up to 36 weeks and decrease following that timepoint	[75]
Sciatic nerve chronic constriction injury	L4-L5 dorsal horn	Increased SHH protein	[79]
	Endoneurial endothelial cells of sciatic nerve	Transient increase in *Shh* mRNA (from 3 to 24 h) Profound downregulation of *Patched-1* and *Gli-1* mRNA lasting for 30 and 60 days, respectively	[18]
	Sciatic nerve (NOS)	*Smoothened* mRNA downregulation between 6 h and 15 days post-injury	Present article
Infra-orbital nerve chronic constriction injury	Endoneurial endothelial cells of infra-orbital nerve	Profound downregulation of *Gli-1* and *Patched-1* mRNAs from 3 to 48 h and 24 h, respectively	[19]
Chronic nerve compression	Sciatic nerve (NOS)	Decreased *dhh* mRNA and protein at 2 weeks post-injury	[59]

^1^ This mention states that the precise cell type was not specified and/or investigated in the study.

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
