# Peer review of "Hedging Against Neuropathic Pain: *Role of Hedgehog Signaling in Pathological Nerve Healing"

_ijms, 2020, doi:10.3390/ijms21239115_

Round 1
Reviewer 1 Report
This review article is interesting and well-written. I have just few considerations and recommendations. In particular, the field of nervous system regeneration invariably drives the attention to the CNS. Therefore, although I understand that this is a review article focusing on the role of Shh pathway in peripheral nerve regeneration, the comparison to the role of Shh in CNS would be very useful to understand at least some of the multiple mechanisms halting the CNS regeneration. In particular, authors should consider the opportunity of extending some of the sections. For example, sections 3.1.1, 3.1.3, 3.1.4, 3.1.6 and 3.1.7 are probably not comprehensive. Moreover, the questions discussed in these sections merit a comparison with the same problems in the CNS. A number of papers have addressed the role of Shh in plasticity, neurotrophic effects, remyelination and so on, in the CNS, and a comparative discussion would increase the interest of the manuscript, in my opinion.
Author Response
We would like to thank the reviewer for his/her support of our work and comments that will allow significant improvement to our manuscript.
We fully agree that the comparison between PNS and CNS regeneration would most probably be of interest to the reader and have thus added additional data in the various aforementioned sections.
As the paper does indeed focus mainly on peripheral nerve regeneration, we have tried to be concise in our additions but have added several sentences in the relevant sections on axonal survival in the CNS (3.1.1), the central neurotrophic effect of SHH in adult neurogenesis, activity-dependent neuroplasticity, reactive astrocytosis following injury, and depression (3.1.4), oligodendrocyte development and remyelination (3.1.7) and provide a parallel between peripheral neuro-glial interactions and similar central interactions illustrated in the context of nigrostriatal circuit maintenance in adulthood (3.2.1). We believe that such concise additions should be sufficient to draw attention to the similarities (and differences) between SHH-mediated peripheral and central nervous system regeneration without overlenghtening and obfuscating the manuscript.
Reviewer 2 Report
This is a concise and timely review about very important subject. This review is nicely written to the readers outside of this research field. Figures are also illustrative.
Author Response
Thank you very much for your interest and support of our work.